# Particle Size Inversion Constrained by *L*_∞_ Norm for Dynamic Light Scattering

**DOI:** 10.3390/ma15207111

**Published:** 2022-10-13

**Authors:** Gaoge Zhang, Zongzheng Wang, Yajing Wang, Jin Shen, Wei Liu, Xiaojun Fu, Changzhi Li

**Affiliations:** School of Electrical and Electronic Engineering, Shandong University of Technology, Zibo 255000, China

**Keywords:** dynamic light scattering, particle sizing, regularization inversion, unimodal distribution, bimodal distribution

## Abstract

Particle size inversion of dynamic light scattering (DLS) is a typically ill-posed problem. Regularization is an effective method to solve the problem. The regularization involves imposing constraints on the fitted autocorrelation function data by adding a norm. The classical regularization inversion for DLS data is constrained by the *L*_2_ norm. In the optimization equation, the norm determines the smoothness and stability of the inversion result, affecting the inversion accuracy. In this paper, the *L*_p_ norm regularization model is constructed. When *p* is 1, 2, 10, 50, 100, 1000, and ∞, respectively, the influence of their norm models on the inversion results of data with different noise levels is studied. The results prove that overall, the inversion distribution errors show a downward trend with the increase of *p*. When *p* is larger than 10, there is no significant difference in distribution error. Compared with *L*_2_, *L*_∞_ can provide better performance for unimodal particles with strong noise, although this does not occur in weak noise cases. Meanwhile, *L*_∞_ has lower sensitivity to noise and better peak resolution, and its inverse particle size distribution is closer to the true distribution for bimodal particles. Thus, *L*_∞_ is more suitable for the inversion of DLS data.

## 1. Introduction

Dynamic light scattering (DLS) is an effective technique to characterize submicron and nanoparticles by measuring particle size and particle size distribution (PSD) [1] which is widely used in fluid dynamics [2], polymer materials [3], medicine [4], biochemistry [5], silica nanoparticles [6], and the ultra-small-angle neutron scattering [7]. The PSD information is obtained by the inversion of the autocorrelation function (ACF) of the scattered light intensity, which mathematically speaking is an ill-posed problem as it involves Fredholm integral equations of the first type. In practice, the existence of noise and rounding errors in ACF data may prevent the existence of solutions, or mean that the solutions are unstable or not unique. A variety of estimation methods have been proposed to accurately estimate PSD from DLS measurement data, including Tikhonov regularization [5], the accumulation method [8], the double exponential method [9], the exponential sampling method [10], the non-negative constrained least squares [11], the CONTIN algorithm [12], and others. Among these algorithms, Tikhonov regularization and the CONTIN based on imposing a regularization constraint on the data fitting terms are the most effective in estimating PSD [5,9,10]. These algorithms use the solutions of a family of well-posed problems that are “adjacent” to the original ill-posed one to approximate the solution of the original problem and produce reasonable results [10]. In this approach, the data-fitting term is used to describe the closeness between the fitting data and the measured data, and the regularization constraint term can make the fitting result more stable and smoother [13]. In classical DLS regularization models, the *L*_2_ norm is used both in the data fitting term and the constraint regularization term [14]. Many studies based on *L*_2_ norm have been carried out to discuss regularization parameter selection, the optimization of ACF baseline, and the information extraction of ACF, and various improvement methods such as Marino’s algorithm have been proposed. However, norms other than *L*_2_ may be employed to improve the accuracy, as well as the stability, of the solution in a variety of applications, such as geophysics [10] and remote-sensing techniques [11]. The resulting DLS inversion is inherently complex and influenced by the different kinds of noise due to fluctuations in the measurement duration, sample turbidity, dust, and the diversity of particle distribution, and so on, causing the inversion results to deviate significantly from the true distribution and ultimately leading to the failure of particle size measurement [1]. Therefore, it is of great significance to study the influence of other norms on the accuracy of inversion. In 2017, Xinjun Zhu [1] studied the influence of the norm of the fitting term on the inversion results based on the *L*_2_ norm regularization, and results have shown that the noise sensitivity of inversion PSD may be improved using *L*_1_ norm. However, the influence of the regularization constraint term on the inversion of DLS data is not yet clear. Based on the classical data fitting term (*L*_2_), the influence of the norm change of regularization constraint term on the inversion results is explored using the CVX toolbox in MATLAB in this paper, and then a DLS inversion method with *L*_∞_ model is proposed. This is more suitable for the inversion of DLS data due to the lower sensitivity to noise and better peak resolution.

## 2. *L*_P_ Norm Constrained Regularization Model for Dynamic Light Scattering

The light-intensity ACF of the scattered light measured in the DLS experiment can be expressed as
(1)$$G^{\left(2\right)}\left(\tau \right)=\langle n\left(t\right)n\left(t+\tau \right)\rangle$$
where *n*(*t*) is the number of photons measured within a certain period *t* and *τ* is the time delay. Angle brackets 〈〉 denote a long-time average. After dividing by the square of the average count value, the normalized ACF is expressed as
(2)$$g^{\left(2\right)}\left(\tau \right)=\frac{G^{\left(2\right)}\left(\tau \right)}{{\langle n(t)\rangle }^{2}}$$

The first-order electric field ACF, $g^{\left(1\right)}\left(\tau \right)$, is obtained by Siegert’s relation, i.e.,
(3)g(1)(τ)=±g(2)(τ)−1β
where *β* is the relevant instrument parameter and ≤1.

The PSD in DLS can be predicted by the electric field ACF, expressed as
(4)$$g^{\left(1\right)}\left(\tau \right)=\int_{0}^{\infty }G\left(\Gamma \right)\exp(-\tau \Gamma )d\Gamma$$
where G(Γ) is the normalized distribution function of the decay constant and $\Gamma =\frac{k_{B}T}{3\pi \eta d}q^{2}$, where *k*_B_ is the Boltzmann constant, *T* is the absolute temperature, *η* is the viscosity of the suspension, *d* is the particle size, $q=\frac{4\pi n_{0}}{\lambda_{0}}\sin\left(\frac{\pi }{2}\right)$ is the scattering vector, *λ*_0_ is the wavelength of the incident beam in a vacuum, *n*_0_ is the refractive index of the suspending medium, and *θ* is the scattering angle.

Equation (4) can be rewritten as
(5)$$g^{\left(1\right)}\left(\tau \right)=\int_{0}^{\infty }f\left(D\right)\exp(-k\tau /D)dD$$
where f(D) is the PSD and *k* can be expressed as
(6)$$k=\frac{k_{B}T}{3\pi \eta }{\left[\frac{4\pi n_{0}}{\lambda_{0}}\sin(\frac{\theta }{2})\right]}^{2}$$

The discrete form of Equation (5) is
(7)$$g^{\left(1\right)}\left(\tau_{j}\right)=\overset{N}{\underset{i=1}{\sum }}\exp(-k\tau_{j}/D_{i})\alpha_{i}$$
where $\alpha_{i}$ is the light scattering amplitude for particles with particle size *D_i_*, which is the mean value of each interval $\left[D_{i},D_{i+1}\right]$ from *N* total intervals, and $\tau_{j}$ is the delay time at the *j*th correlator channel.

Equation (7) may be written as a linear equation
(8)g=Ax
where g is a vector with elements $g^{\left(1\right)}\left(\tau_{j}\right)$, A is a matrix with elements exp(−*kτ_j_*/*D_j_*), and *x* is a vector with elements *x_i_*.

The solution of Equation (8) can be expressed as the following least-square problem
(9)$$∥Ax-g{∥}_{2}=\min=x_{LS}$$
which, however, is an ill-posed problem. If the data g is noisy, the solution *x* may have large random fluctuations and present instability. The Tikhonov regularization method can effectively solve such ill-posed problems [9]. In the classical Tikhonov method, the *L*_2_ norm regularization term $∥Lx{∥}_{2}$ is introduced into Equation (9) to constrain. Thus, the approximate stable solution of Equation (9) is obtained through the optimization of the following Equation (10) [12]
(10)$$M_{1}^{\alpha }\left(x\right)=∥Ax-g{∥}_{2}+\alpha ∥Lx{∥}_{2}\begin{array}{cc} s.t. & x\geq 0 \end{array}$$
where *α* is the regularization parameter, which controls the accuracy and stability of the ill-posed solution,∥∥
is the Euclidean norm, and *L* is the regularization matrix. Considering the actual physical meaning of the PSD, *x* is required to be non-negative. To study the influence of the regularization constraint norm on the inversion results, we consider the following optimization problem
(11)$$M_{2}^{\alpha }\left(x\right)=∥Ax-g{∥}_{2}+\alpha ∥Lx{∥}_{p}\begin{array}{cc} s.t. & x\geq 0 \end{array}$$
where $∥{∥}_{p}$ denotes the *L*_p_ norm. If we introduce *y* = *Lx*, the *L**_p_* norm of the penalty function *Lx* is $∥Lx{∥}_{p}$ where
(12)$${∥y∥}_{p}={\left(\overset{n}{\underset{i}{\sum }}y_{i}^{p}\right)}^{\frac{1}{p}}$$

In Equation (12), when *p* takes ∞, the *L**_p_* norm is the infinite norm(13)$${∥y∥}_{\infty }=\underset{1\leq i\leq n}{max}\left|y_{i}\right|$$

Theoretically, *p* can be set to a value from 0 to ∞. However, the relevant fitting data is non-convex when *p* < 1, and thus we consider *p* in the range [1, ∞). We will choose *p* = 1, 2, 10, 50, 100, 1000, and ∞ to investigate the differences between different models. The principle of the process of fitting data using the *L*_∞_ norm is different compared with other norms. In the data-fitting process, the fit result obtained by the *L*_∞_ norm is the mid-range number of the series, i.e., the average of the maximum and minimum values, when the principle of the fitting is the mid-range number regression. These data-fitting results are highly sensitive to outliers and have a maximum positive deviation and a minimum negative deviation of the sum of zero. The use of the *L*_∞_ norm model in the data-fitting process can maximize the distance value of the corresponding point in the data with the smallest absolute distance from the upper limit function of the distance specification and to ensure that all points in the final solution set are satisfactory, so as to improve the noise immunity of the model [15]. Data fitting at different parameters can be achieved by entering the relevant simulation data in MATLAB and using the CVX toolbox.

## 3. Simulation Experiment Parameters and Evaluation Indicators

To conveniently explore the influence of different norms regularization terms on the inversion results, it is necessary to generate a variety of ACF data to simulate the true experimental environment. In the simulation inversion, simulated noisy intensity ACF data are obtained by the following expression
(14)$$G_{noise}\left(\tau_{j}\right)=G\left(\tau_{j}\right)+\delta n\left(\tau_{j}\right)$$
where $G_{noise}\left(\tau_{j}\right)$ denotes light intensity ACF with noise, $G\left(\tau_{j}\right)$ is the corresponding ACF without noise, $n\left(\tau_{j}\right)$ denotes Gaussian noise, *δ* is the noise level, and *j* = 1, 2,..., *M*, *M* is the number of channels of the digital correlator.

The various PSDs are obtained by the Johnson SB function [16], which is given by
(15)$$f\left(d\right)=\frac{\sigma }{\sqrt{2\pi }}{\left[t\left(1-t\right)\right]}^{-1}exp\left\{-0.5{\left[\mu +\sigma ln\left(\frac{t}{1-t}\right)\right]}^{2}\right\}$$

In Equation (15), $t=\left(d-d_{min}\right)/\left(d_{max}-d_{min}\right)$ is the normalized value of particle size, 0≤t≤1, $d_{min}$ is the smallest particle size, $d_{max}$ is the maximum particle size, and *μ* and *σ* are the distribution parameters. Different forms of PSD can be obtained by changing the distribution parameters, whereas bimodal particles may be obtained by mixing two unimodal particles with different intensity ratios.

Simulated experiments have been performed in the following conditions: the wavelength of the laser *λ* in a vacuum is 632.8 nm, the refractive index *n* of the solution is 1.33, the absolute temperature *T* is 273 K, the Boltzmann constant *k*_B_ = 1.3807 × 10^−23^ J/K, the viscosity coefficient *η* = 0.89 × 10^−3^ Pa·s, and the scattering angle *θ* = 90°.

To compare the inversion accuracy of PSDs under different norms models, the following parameters are introduced as performance parameters:(1)Distribution error *E.*

The distribution error *E* is expressed as
(16)$$E=∥f\left(d_{i}\right)-f_{1}{(d_{i})∥}_{2}/∥f_{1}{(d_{i})∥}_{2}$$
where $f\left(d_{i}\right)$ is the PSD simulated by the Johnson-SB function and $f_{1}\left(d_{i}\right)$ is the result of the inversion.


(2)Peak value error $E_{PV}$


The peak value error $E_{PV}$ is expressed as
(17)$$E_{PV}=\left(\left|d_{ps}-d_{1ps}\right|/d_{ps}\right)\times 100\%$$
where $d_{ps}$ is the inversion peak value and $d_{1ps}$ is the peak value of the simulated PSD (i.e., true peak).

The $E_{PV}$ and *E* obtained by the inversion are used to compare the effects of different norms. The closer the peak value to the true value, the smaller the distribution error, and thus the better the PSD obtained by the inversion. The regularization scheme is also related to the regularization parameters. To obtain better inversion results, it is necessary to choose appropriate regularization parameters. Generally, there are many criteria for the selection of regularization parameters, such as GCV [17], L-curve [18], and others. Considering that the L-curve is more robust against noise and provides better stability than other criteria, L-curve is used to select the regularization parameters in this paper.

## 4. Analysis of Simulated Data with Different Norms

### 4.1. Simulated Inversion with Different Norms

To study the inversion effect of the model (11) under the different norms, unimodal particles and bimodal particles that are widely used in industry were used as examples in this paper. We have chosen 381.1 nm, 376.3 nm/685.6 nm, and 236.5 nm/685.6 nm particles for the simulated inversion. According to the Equation (15), the range and location of the PSD is controlled by the simulation data parameters *μ* and *σ*, which can be adjusted to obtain the right range and position of the peak. Table 1 summarizes the relevant simulation experiment parameters for the above particle systems.

We consider the inversion of the ACF data for three particle systems with different noise levels: 0, 0.0005, 0.001, 0.005, 0.01 at *p* = 1, 2, 5, 10, 100, 1000, and ∞, respectively. The inversion results and their distribution errors and peak value errors are shown in Figure 1, Figure 2 and Figure 3 and Table 2, Table 3, Table 4, Table 5 and Table 6. For the 381.1 nm unimodal particles, the inversion data table is not given because the inversion peak value error is zero for the different norms and noise levels.

The simulation results in Figure 1 and Table 2 show that: (1) For the unimodal particles, the peak value error is 0, independently of the used norm and noise level. (2) The inversion results of the *L*_1_ are poorly smoothed when the noise level is increased from 0 to 0.01. The distribution error of *L*_1_ is the largest compared with the other norms, with a maximum value of 0.1457. (3) The difference between the distribution errors of *L*_2_ and *L*_10_~*L*_∞_ is not obvious at weak noise levels of 0, 0.0005, 0.001, and 0.005 with a maximum difference of 0.0203. At the strong noise level of 0.01, the distribution errors of *L*_10_~*L*_∞_ have no big difference, while the distribution error of *L*_2_ is 0.1309, which is second only to *L*_1_ in distribution errors.

It can be seen from the results reported in Figure 2 and Figure 3 and Table 3, Table 4, Table 5 and Table 6 that: (1) For both particle systems, the inversion results differ greatly for different norms at the same noise level. The gap between the inversion PSD of *L*_1_ and the true distribution is the largest, followed by the *L*_2_. The difference between the inversion results of other norms and the true distribution is not significant. (2) With the increase of the *p* parameter, the distribution error shows a decreasing trend, and the peak value error remains unchanged or decreases. (3) Compared with the distribution error of *L*_1_ and *L*_2_, the maximum reduction of the distribution errors for *L*_∞_ are 0.3597, 0.2613, and 0.3587, 0.1522 for the 376.3 nm/685.6 nm and 236.5 nm/685.6 nm particle systems, respectively. (4) For both particle systems, the peak value error is not discussed here because the inversion results of *L*_1_ are poorly smoothed, sometimes the bimodal PSD characteristics cannot be obtained, and the gap with the true PSD is very large. Compared with the peak value error of *L*_2_, the maximum reduction in peak value errors for *L*_∞_ are 0.0263/0.0728 and 0.0432/0 for the 376.3 nm/685.6 nm and 236.5 nm/685.6 nm particle systems, respectively.

It can be concluded from the above analysis that: (1) The difference between the PSD obtained with *L*_1_ and the true PSD is the largest. The difference between the distribution errors of *L*_10_~*L*_∞_ is not obvious under various noise conditions. At the strong noise level, the distribution errors of *L*_10_~*L*_∞_ are smaller than that of *L*_2_. In general, the distribution errors of the inversions show a decreasing trend with increasing *p*. After *p* becomes larger than 10, the distribution errors of each norm do not differ significantly. (2) In practical applications, considering the simple calculation of *L*_∞_, its distribution error and peak value error are smaller compared with the *L*_2_. *L*_∞_-based regularization, and is also more effective against noise and concerning bimodal resolution. We conclude that the *L*_∞_ norm model provides better performance than the classical *L*_2_ norm model, and furthermore, it is more suitable for DLS inversion.

### 4.2. Discussion

The following phenomena can be summarized from Figure 1, Figure 2 and Figure 3 and Table 2, Table 3, Table 4, Table 5 and Table 6: (1) The difference between the distribution errors of *L*_10_~*L*_∞_ is not obvious under various noise conditions. (2) In general, the distribution error of *L*_10_~*L*_∞_ is smaller compared with *L*_2_. The reason for this phenomenon can be explained by the principle of norms. The unit balls defined under the *p* norms are all convex sets, but when 0 < *p* < 1, the unit ball under that definition is not a convex set. Figure 4 shows the shape of the unit ball for different values of *p*, from 0.25 to ∞. It can be seen from Figure 4 that for *L*_10_~*L*_∞_, their unit ball shapes are closer, and therefore the results obtained from *L*_10_~*L*_∞_ are also closer. Considering from the equation (13) that *L*_∞_ has the features of a simple calculation, *L*_∞_ is chosen for the inversion.

## 5. Inversion Analysis of *L*_∞_ Model

To assess the performance of the *L*_∞_ and *L*_2_ models in more detail, other systems with different PSD have been analyzed. In particular, we consider systems with a narrow unimodal PSD at 150.7 nm, a broad unimodal PSD at 390.1 nm, and several bimodal PSDs with peak value position ratios of 1.41, 1.82, and 3.06 at 466.1 nm/655.7 nm, 376.3 nm/685.6 nm, and 246.5 nm/755.5 nm, respectively.

### 5.1. Unimodal Particles

The shape distribution parameters of the unimodal particles with broad and narrow PSDs are given in Table 7. Figure 4 and Figure 5 show the comparison of the inversion results inverted by the *L*_2_ and *L*_∞_ model for the two particle systems and different noise levels. Table 8 shows the peak value positions of the inversion results. Table 9 shows the distribution errors of the inversion results for different noise levels and *p* values.

It can be concluded from Figure 5 and Figure 6 and Table 8 and Table 9 that: (1) For a noise level in the range 0~0.005, the distribution errors of the *L*_2_ model for the two distributions are slightly lower than the distribution errors of *L*_∞_, at most 0.0203, 0.0243 lower, respectively. (2) For a noise level equal to 0.01, the distribution error of the *L*_∞_ norm is lower than that of the *L*_2_ norm model (by 0.0342 and 0.0162, respectively). (3) The peak value errors from the *L*_∞_ and *L*_2_ models are the same almost independently of the noise. Overall, we conclude that the *L*_∞_ model for unimodal particles has clear advantages in the presence of strong noise, while the advantage is not obvious for weak noise.

### 5.2. Bimodal Particles

Three sets of simulated PSD data are used to generate the ACF for bimodal particles. The shape distribution parameters for the distributions with parameters 466.1 nm/655.7 nm, 376.3 nm/685.6 nm, and 246.5 nm/755.5 nm, respectively, are given in Table 10. Figure 6, Figure 7 and Figure 8 show the results obtained with the *L*_2_ and *L*_∞_ models for the same PSD and different noise levels. Table 11 shows the peak value positions of the inversion results. Table 12 shows the distribution errors inversed by the two models under different noise levels:

It can be concluded from Figure 7, Figure 8 and Figure 9 and Table 11 and Table 12 that: (1) For 466.1 nm/655.7 nm bimodal particles with a peak value position ratio of 1.41, the *L*_∞_ model leads to the effective reconstruction of the bimodal PSD for the various noise levels, whereas *L*_2_ can obtain a clear bimodal feature only when the noise level is 0 or 0.0005. The bimodal feature is instead nearly lost for other noise levels. The peak value errors and distribution errors of *L*_∞_ are 0.0214/0.1064, 0.279 lower than those of *L*_2_. (2) For the bimodal particles of 376.3 nm/685.6 nm (peak value ratio 1.82) and 246.5 nm/755.5 nm (peak value ratio 3.06), although the bimodal PSD can be recovered by using both models, they are not robust against noise. As the noise level increases, the inversion PSD with *L*_2_ broadens, and the peak values errors and distribution errors are larger than those obtained with *L*_∞_. The inversion distribution errors for the 376.3 nm/685.6 nm and 246.5 nm/755.5 nm particles are at most 0.2623 and 0.123 higher, respectively. The peak value errors are at most 0.0263/0.0582 and 0.0102/0.0113 higher, respectively. (3) In general, the *L*_∞_ model is less sensitive to noise than the *L*_2_ model under noise levels equal to 0, 0.0005, 0.001, 0.005, and 0.01. Compared with the *L*_2_ model, the inversion peak value position and peak value height obtained by the *L*_∞_ model are closer to the true values, and the inversion distribution errors are lower. For a bimodal PSD with a low peak value position ratio, the *L*_∞_ model shows a stronger bimodal resolution.

### 5.3. Discussion

The following phenomena can be summarized from Figure 5, Figure 6, Figure 7, Figure 8 and Figure 9 and Table 8, Table 9, Table 11 and Table 12, that: The inversion peak value position and peak value height obtained by the *L*_∞_ are closer to the true values, and the inversion distribution errors are lower than the *L*_2_. The reason for this phenomenon can be explained as that for *L*_2_ and *L*_∞_, both are essentially based on the principle of looking for a solution to the DLS regularized problem, which is essentially an iterative solution problem to Equations (10) and (18)
(18)$$M_{3}^{\alpha }\left(x\right)=∥Ax-g{∥}_{2}+\alpha ∥Lx{∥}_{\infty }\begin{array}{cc} s.t. & x\geq 0 \end{array}$$

The three adjacent units in the x direction are taken, and they are set to correspond to three elements *x*_1_, *x*_2_, and *x*_3_, respectively, and the two parameter subspaces are defined as follows
(19)$$S^{2}=span\left\{\left[x_{1},0\right],\left[0,x_{3}\right]\right\}$$

The contour lines of *L*_2_ and *L*_∞_ in subspace *S*^2^ are shown in Figure 10, and the minimum value in the space *S*^2^ is located at the origin (*x*_2_, *x*_2_) when *x*_2_ takes a fixed value.

For the discussion’s convenience, let us define the sets *A*_1_ and *A*_2_ as
(20)$$\{\begin{array}{c} A_{1}=\left\{x|x={\left[x_{1},x_{2},x_{3},\cdots \right]}^{T},\right\}\\ x_{1}=x_{3}\\ A_{2}=\left\{x|x={\left[x_{1},x_{2},x_{3},\cdots \right]}^{T},\right\}\\ x_{1}=x_{2}=x_{3} \end{array}$$

In Equation (20), *x* represents the model vector, and each element in *x* represents the parameters in the solution space. The point (*x*_2_, *x*_2_) is the origin in the solution space, the set *A*_1_ corresponds to the two coordinate axes of the origin in the solution space, and the set *A*_2_ corresponds to the origin (*x*_2_, *x*_2_). When using the iterative method to find the minimum value of $M_{1}^{\alpha }\left(x\right)$, the fastest descent direction of the *L*_2_ regularization term must point to the origin, therefore *x*_k+1_ is more likely to fall within the set *A*_2_ and its neighborhood. When using the iterative method to find the minimum value of $M_{3}^{\alpha }\left(x\right)$, the fastest descent direction of the *L*_∞_ regularization term deviates from the origin and tends to the dotted line position in Figure 10b. In this condition, *x*_k+1_ tends to fall within the set *A*_1_ and its neighborhood. The *L*_∞_ model will make the spatial gradient vector ***x*** more likely to obtain a sparse solution when the whole solution space is considered. The regularization constraint is also more resistant to noise interference, making the obtained solution more stable. In addition, according to Equation (10), the squaring operation in the *L*_2_ model optimization requires at least quadratic programming to solve, whereas the computation of *L*_∞_ is simpler and easier to implement. Therefore, we conclude that for the data inversion of DLS bimodal particles and strong noises, their solutions are more complex. The *L*_∞_ model with simple calculation and easier to obtain sparse solution has better noise resistance and robustness.

## 6. Experimental Data Inversion Analysis

Experimental data were obtained from a vertically polarized He-Ne laser at a wavelength of 632.8 nm and a 72-channel photon correlator with *θ* = 90° (Malvern Zetasizer Nano ZS90). The tested samples were unimodal particles (61 ± 4 nm) (Duke Scientific catalog no. 3030A), (203 ± 5 nm) (Duke Scientific catalog no. 3200A) and bimodal particles (31 ± 3 nm)/(203 ± 5 nm) made up of standard polystyrene lattices (Duke Scientific Beijing China) with nominal diameters of 60 and 200 nm (Duke Scientific catalog no. 3060A and 3200A). The samples were prepared by adding standard particles to deionized water and subjecting them to 5 min of ultrasonication. The DLS measurements were carried out by placing the sample under test in a 12 mm diameter high quality stereo quartz cuvette at a constant temperature of 298.15 K. The *L*_2_ and *L*_∞_ were used, respectively, to invert the experimental data. The inversion results are shown in Figure 11 and Table 13. To make the experimental data more convincing, the above-mentioned particles were tested using the Malvern instrument (Malvern Zetasizer Nano ZS90), and the results and associated data are shown in Figure 12 and Table 14.

Looking at the inversion results and their peak value locations and errors in Table 13 and Table 14, it can be concluded that: (1) *L*_∞_ leads to the more accurate PSD and a more accurate peak value position than the standard inversion methods and the *L*_2_. (2) The peak value errors of *L*_2_ and *L*_∞_ are 0.85% and 0.51% for the 61 nm unimodal particles. For the inversion of 31 nm/203 nm bimodal particles, the peak position error of the *L*_∞_ norm model is 1.19%/0.99%, which is more accurate than that of *L*_2_. The inversion results of *L*_∞_ are closer to the true values than those of the Malvern instrument. Based on the above conclusions, it can be seen that the *L*_∞_ also performs well in the inversion of the experimental data, with smaller inversion errors and better performance than the *L*_2_.

## 7. Conclusions

The existence of noise in DLS technology reduces the reliability of PSD obtained from ACF data. In this paper, the *L*_P_ norm constrained regularization model is constructed, and the influence of different *p*-norm regularization, which *p* is 1, 2, 10, 50, 100, 1000, and ∞, respectively, on the inversion results has been studied by using simulated data. It is concluded that the inversion distribution error decreases with the increase of *p*, and that there is no significant difference in the distribution error when *p* is larger than 10. On this basis, the *L*_∞_ model has the advantages of simple calculation and easy-to-obtain sparse solutions, making the obtained solutions more robust and improving the noise resistance. A DLS inversion method constrained by *L*_∞_ norm regularization has been proposed. Using the *L*_2_ and *L*_∞_, the simulated data of unimodal and bimodal particles with different noise levels were studied. We conclude that, compared with the classical *L*_2_ model, *L*_∞_ is more advantageous in the case of strong noise and not obvious in the case of weak noise for the inversion of unimodal particles. For bimodal particles, *L*_∞_ has low sensitivity to noise and a strong bimodal resolution, and its inversion result is closer to the true PSD. For experiment particles with 61 nm unimodal PSD or 31 nm/203 nm bimodal PSD, the peak value errors are 0.51%, and 1.19%/0.99%, respectively. The results of the measured data also confirm the conclusions of the simulated data. The results of the simulation and experimental data analysis show that the *L*_∞_ model provides an effective method to improve the accuracy of DLS measurements. The *L*_∞_ model has a strong resistance to noise and can be considered for solutions in the case of larger noise, such as ultra-low concentration measurements and online measurements of flow processes.

## Figures and Tables

**Figure 1 materials-15-07111-f001:**
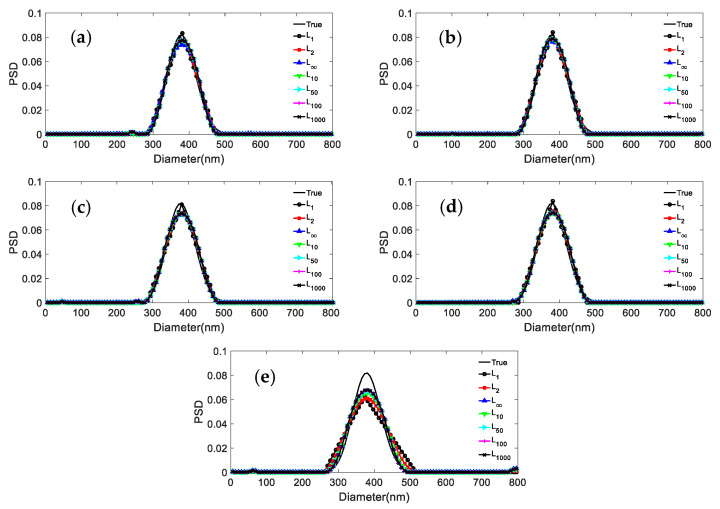
Inversion PSD of 381.1 nm unimodal particles under different norms and noise levels. (**a**) 0; (**b**) 0.0005; (**c**) 0.001; (**d**) 0.005; (**e**) 0.01.

**Figure 2 materials-15-07111-f002:**
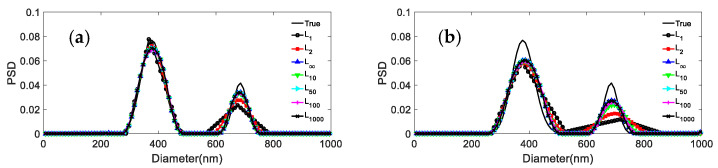
Inversion PSD of 376.3 nm/685.6 nm bimodal particles under different norms and noise levels. (**a**) 0; (**b**) 0.0005; (**c**) 0.001; (**d**) 0.005; (**e**) 0.01.

**Figure 3 materials-15-07111-f003:**
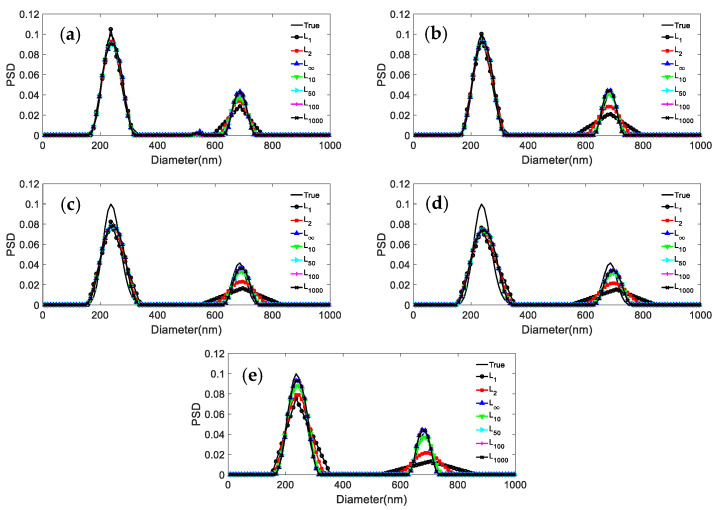
Inversion PSD of 236.5 nm/685.6 nm bimodal particles under different norms and noise levels. (**a**) 0; (**b**) 0.0005; (**c**) 0.001; (**d**) 0.005; (**e**) 0.01.

**Figure 4 materials-15-07111-f004:**
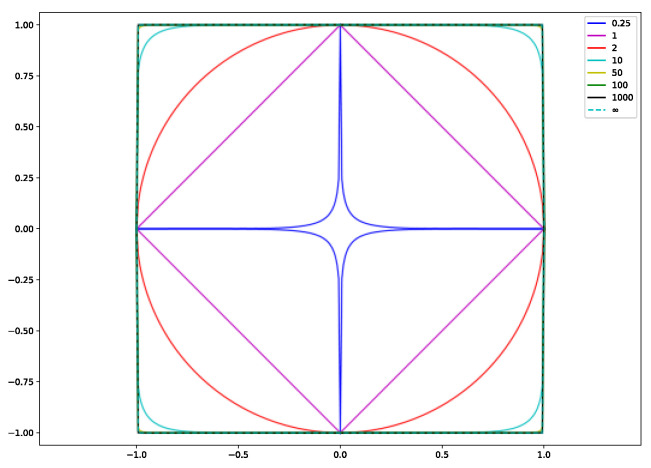
The shape of the unit ball for different values of *p* from 0.25 to ∞.

**Figure 5 materials-15-07111-f005:**
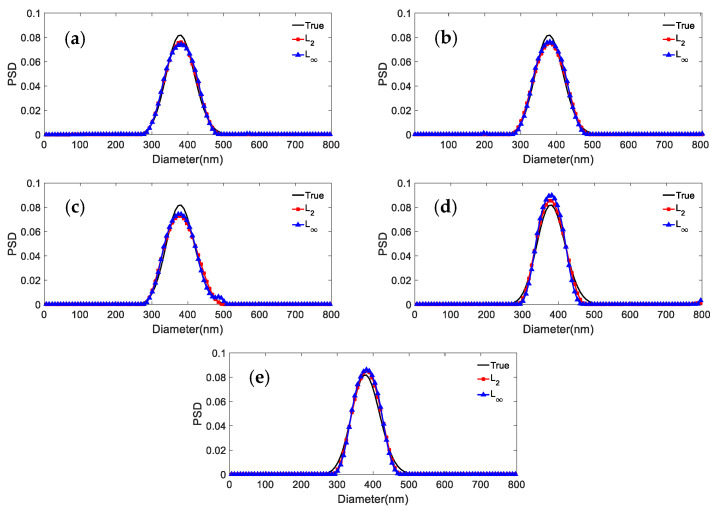
Inversion PSD of 390.1 nm unimodal particles under different norms and noise levels. (**a**) 0; (**b**) 0.0005; (**c**) 0.001; (**d**) 0.005; (**e**) 0.01.

**Figure 6 materials-15-07111-f006:**
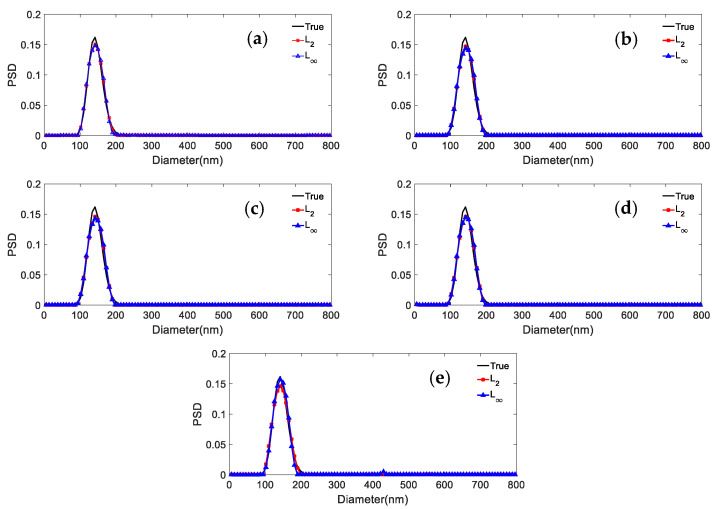
Inversion PSD of 150.7 nm unimodal particles under different norms and noise levels. (**a**) 0; (**b**) 0.0005; (**c**) 0.001; (**d**) 0.005; (**e**) 0.01.

**Figure 7 materials-15-07111-f007:**
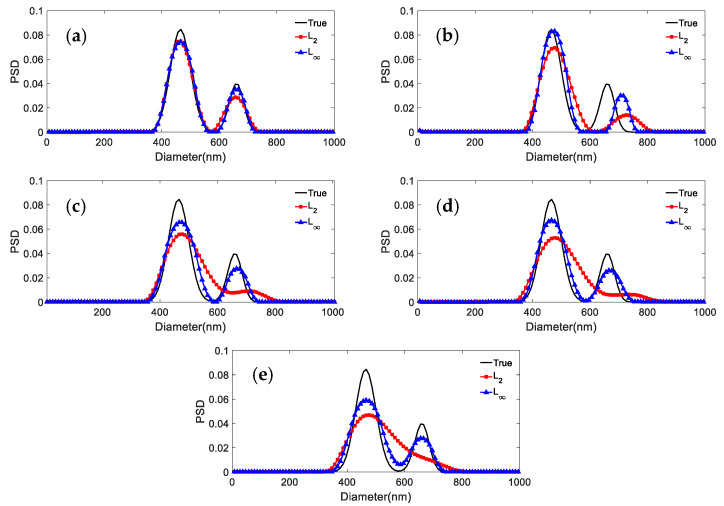
Inversion PSD of 466.1 nm/655.7 nm bimodal particles under different noise levels. (**a**) 0; (**b**) 0.0005; (**c**) 0.001; (**d**) 0.005; (**e**) 0.01.

**Figure 8 materials-15-07111-f008:**
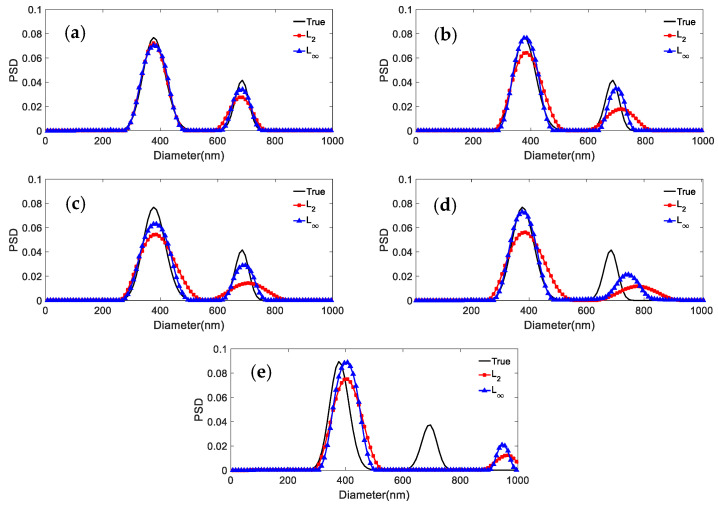
Inversion PSD of 376.3 nm/685.6 nm bimodal particles under different noise levels. (**a**) 0; (**b**) 0.0005; (**c**) 0.001; (**d**) 0.005; (**e**) 0.01.

**Figure 9 materials-15-07111-f009:**
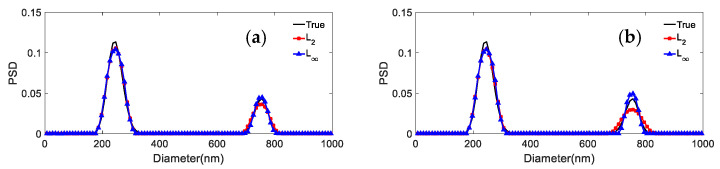
Inversion PSD of 246.5 nm/755.5 nm bimodal particles under different noise levels. (**a**) 0; (**b**) 0.0005; (**c**) 0.001; (**d**) 0.005; (**e**) 0.01.

**Figure 10 materials-15-07111-f010:**
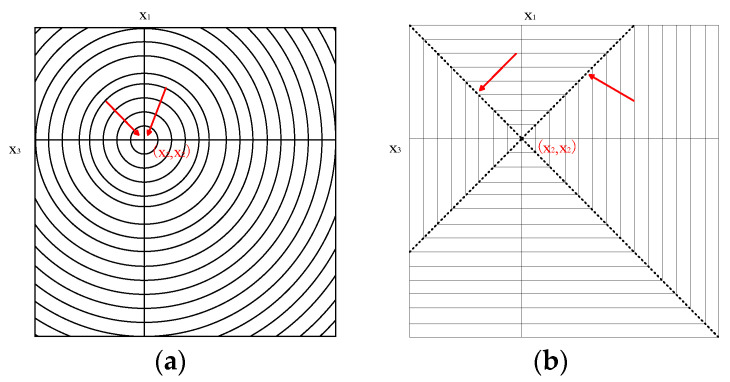
Schematic diagram of the *L*_2_ and *L*_∞_ regular functions of the solution space. (**a**) *L*_2_ regularization; (**b**) *L*_∞_ regularization.

**Figure 11 materials-15-07111-f011:**
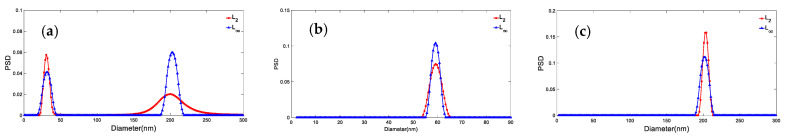
Bimodal and unimodal particles inversion results of *L*_2_ and *L*_∞_. (**a**) 31/203 nm; (**b**) 61 nm; (**c**) 203 nm.

**Figure 12 materials-15-07111-f012:**
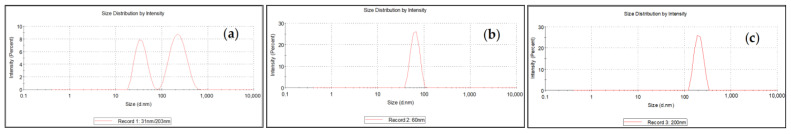
Bimodal and unimodal particles inversion results from the Malvern instrument (**a**) 31 nm/203 nm; (**b**) 61 nm; (**c**) 203 nm.

**Table 1 materials-15-07111-t001:** Simulation data parameters for unimodal and bimodal particles.

Particles	Parameters					
	*D*_min_/nm	*D*_max_/nm	*μ* _1_	*μ* _2_	*σ* _1_	*σ* _2_
381.1 nm	10	1000	3.0	-	9.0	-
376.3 nm/685.6 nm	10	1000	3.0	−3.0	7.0	7.0
236.5 nm/685.6 nm	10	1000	7.0	−3.0	7.0	7.0

**Table 2 materials-15-07111-t002:** Distribution errors data of 381.1 nm unimodal particles under different norms and noise levels.

Noise	*L* _1_	*L* _2_	*L* _10_	*L* _50_	*L* _100_	*L* _1000_	*L* _∞_
0	0.0924	0.0690	0.0729	0.0756	0.0759	0.0761	0.0893
0.0005	0.0931	0.0702	0.0776	0.0774	0.0774	0.0772	0.0817
0.001	0.1005	0.0876	0.0960	0.1026	0.1005	0.1008	0.1033
0.005	0.1236	0.0987	0.0998	0.1021	0.1081	0.1104	0.1093
0.01	0.1457	0.1309	0.1002	0.1095	0.1109	0.1181	0.1196

**Table 3 materials-15-07111-t003:** Distribution errors data of 376.3 nm/685.6 nm bimodal particles under different norms and noise levels.

Noise	*L* _1_	*L* _2_	*L* _10_	*L* _50_	*L* _100_	*L* _1000_	*L* _∞_
0	0.3833	0.1764	0.1346	0.2254	0.2251	0.1250	0.1251
0.0005	0.4986	0.3961	0.2049	0.2073	0.3211	0.1689	0.1688
0.001	0.5592	0.4608	0.2067	0.1989	0.2004	0.1987	0.1995
0.005	0.5477	0.4826	0.2904	0.2958	0.2956	0.2442	0.2455
0.01	0.5897	0.5935	0.5650	0.5554	0.5542	0.5514	0.5451

**Table 4 materials-15-07111-t004:** Distribution errors data of 236.5 nm/685.6 nm bimodal particles under different norms and noise levels.

Noise	*L* _1_	*L* _2_	*L* _10_	*L* _50_	*L* _100_	*L* _1000_	*L* _∞_
0	0.2215	0.1491	0.1086	0.1052	0.1048	0.1049	0.1048
0.0005	0.3762	0.2695	0.1789	0.1601	0.1486	0.1557	0.1556
0.001	0.5129	0.3679	0.2145	0.2375	0.2239	0.2345	0.2343
0.005	0.6025	0.3853	0.2552	0.2399	0.2358	0.2438	0.2438
0.01	0.5964	0.5745	0.4738	0.4466	0.4333	0.4228	0.4223

**Table 5 materials-15-07111-t005:** Peak value errors of 376.3 nm/685.6 nm bimodal particles under different norms and noise levels.

Noise	*L*_1_ (%)	*L*_2_ (%)	*L*_10_ (%)	*L*_50_ (%)	*L*_100_ (%)	*L*_1000_ (%)	*L*_∞_ (%)
0	2.65/1.46	0/1.46	0	0	0	0	0
0.0005	0/1.46	0/1.46	0/1.46	2.63/0	0/1.46	2.63/0	2.63/0
0.001	-	0/1.46	2.63/1.46	0/1.46	0/1.46	0/1.46	0/1.46
0.005	-	0	0	0	2.63/0	0	0
0.01	-	2.63/7.28	0/4.38	0	0	0	0

**Table 6 materials-15-07111-t006:** Peak value errors of 236.5 nm/685.6 nm bimodal particles under different norms and noise levels.

Noise	*L*_1_ (%)	*L*_2_ (%)	*L*_10_ (%)	*L*_50_ (%)	*L*_100_ (%)	*L*_1000_ (%)	*L*_∞_ (%)
0	0	0	0	0	0	0	0
0.0005	0	0	0	0	0	0	0
0.001	0/1.46	4.32/1.46	4.32/0	4.32/0	4.32/0	4.32/0	4.32/0
0.005	2.92/0	4.32/1.46	4.32/1.46	0/1.46	0/1.46	0/1.46	0/1.46
0.01	0/2.92	4.32/1.46	0/1.46	4.32/1.46	4.32/1.46	4.32/1.46	4.32/1.46

**Table 7 materials-15-07111-t007:** Simulation data parameters for unimodal particles.

Particles	Parameters			
	*D*_min_/nm	*D*_max_/nm	*μ*	*σ*
390.1 nm	10	1000	1.0	9.0
150.7 nm	10	1000	3.0	9.0

**Table 8 materials-15-07111-t008:** Peak values and its errors data of unimodal particles inverted by *L*_2_ and *L*_∞_ models under different noise levels.

Particles	Noise	*L* _2_		*L* _∞_	
		Peak Value Position/nm	Peak Value Error (%)	Peak Value Position/nm	Peak Value Error (%)
390.1 nm	0	390.1	0	390.1	0
	0.0005	390.1	0	390.1	0
	0.001	390.1	0	390.1	0
	0.005	390.1	0	390.1	0
	0.01	373.1	4.36	373.1	4.36
150.7 nm	0	150.7	0	150.7	0
	0.0005	150.7	0	150.7	0
	0.001	150.7	0	150.7	0
	0.005	150.7	0	150.7	0
	0.01	150.7	0	150.7	0

**Table 9 materials-15-07111-t009:** Distribution errors data of unimodal particles inverted by *L*_2_ and *L*_∞_ models under different noise levels.

Noise	390.1 nm		150.7 nm	
	*L* _2_	*L* _∞_	*L* _2_	*L* _∞_
0	0.0690	0.0893	0.0751	0.0994
0.0005	0.0735	0.0928	0.0937	0.1062
0.001	0.0603	0.0787	0.0848	0.1061
0.005	0.0894	0.0962	0.0823	0.1041
0.01	0.1994	0.1652	0.1505	0.1343

**Table 10 materials-15-07111-t010:** Simulation data parameters for bimodal particles.

Particles	Parameters					
	*D*_min_/nm	*D*_max_/nm	*μ* _1_	*μ* _2_	*σ* _1_	*σ* _2_
466.1 nm/655.7 nm	10	1000	1.0	−3.0	7.0	7.0
376.3 nm/685.6 nm	10	1000	3.0	−3.0	7.0	7.0
246.5 nm/755.5 nm	10	1000	7.0	−7.0	7.0	9.0

**Table 11 materials-15-07111-t011:** Peak values and their errors data of bimodal particles inverted by *L*_2_ and *L*_∞_ models under different noise levels.

Particles	Noise	*L* _2_		*L* _∞_	
		Peak Value Position/nm	Peak Value Error (%)	Peak Value Position/nm	Peak Value Error (%)
466.1 nm/655.7 nm	0	466.1/655.7	0	466.1/655.7	0
	0.0005	476.1/705.6	2.14/7.61	466.1/665.7	0/1.53
	0.001	476.1/735.5	2.14/12.17	466.1/665.7	0/1.53
	0.005	476.1/-	2.14/-	466.1/655.7	0/1.53
	0.01	476.1/-	2.14/-	466.1/655.7	0/1.53
376.3 nm/685.6 nm	0	376.3/685.6	0	376.3/685.6	0
	0.0005	386.2/715.6	2.63/4.38	376.3/695.6	0/1.46
	0.001	379.4/705.6		376.3/695.6	
	0.005	386.2/785.4	2.63/14.56	376.3/745.5	0/8.74
	0.01	406.2/965.1	7.95/40.77	406.2/945.1	7.95/37.85
246.5 nm/755.5 nm	0	246.5/755.5	0	246.5/755.5	0
	0.0005	246.5/755.5	0	246.5/755.5	0
	0.001	246.5/755.5	0	246.5/755.5	0
	0.005	246.5/765.5	0	246.5/765.5	0
	0.01	246.5/755.5	1.02/2.45	246.5/755.5	0/1.32

**Table 12 materials-15-07111-t012:** Distribution errors data of bimodal particles inverted by *L*_2_ and *L*_∞_ models under different noise levels.

Noise	466.1 nm/655.7 nm		376.3 nm/685.6 nm		246.5 nm/755.5 nm	
	*L* _2_	*L* _∞_	*L* _2_	*L* _∞_	*L* _2_	*L* _∞_
0	0.1746	0.1251	0.1491	0.1048	0.1168	0.0968
0.0005	0.4565	0.2470	0.3610	0.1839	0.1952	0.1106
0.001	0.5553	0.3113	0.3588	0.0965	0.2326	0.1096
0.005	0.5327	0.2537	0.5410	0.3556	0.3876	0.3030
0.01	0.5531	0.4798	0.6218	0.5933	0.2689	0.1616

**Table 13 materials-15-07111-t013:** Peak values and their errors of bimodal and unimodal particles inverted by *L*_2_ and *L*_∞_ models.

Particles	*L* _2_		*L* _∞_	
	Peak Value Position/nm	Peak Value Error (%)	Peak Value Position/nm	Peak Value Error (%)
31 nm/203 nm	30.63/198.7	1.19/2.12	30.63/205	1.19/0.99
61 nm	60.48	0.85	60.69	0.51
203 nm	205	0.98	202.5	0.25

**Table 14 materials-15-07111-t014:** Peak values position of standard inversion.

Particles	Peak Value Position/nm
31 nm/203 nm	36.58 nm/241.5 nm
61 nm	65.2 nm
203 nm	207.0 nm

## Data Availability

Data are contained within the article. The data presented in this study can be requested from the authors.

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
