# Peer review of "Particle Size Inversion Constrained by L Norm for Dynamic Light Scattering"

_materials, 2022, doi:10.3390/ma15207111_

Round 1

Reviewer 1 Report

The authors present a size inversion method for dynamic light scattering (DLS) by Lp norm regularization. This is a mathematically ill-posed problem, as the authors acknowledge, From the experimentalist's point of view it is important to have methodologies that perform this inversion accurately.  For the work to be useful, however, the authors should consider the following points.  

Performance against simulated data is of course needed to establish the general qualities of the method but the true test is against real data. The authors present both, but tests against simulated data are excessive while tests against real data are insufficient. Tests against simulated data can and should be done in fewer than 8 figures.

With respect to experimental data (Fig 10, Table 13), they show the inverted distributions for L2 and L∞ and compare these to the nominal size of commercial unimodal particles. There is now such a thing as "unimodal" when it comes to real particles. The authors should determine the distributions experimentally by TEM. They should also present comparisons against standard inversion methods such CONTIN or bimodal models, which are typically distributed with light scattering equipment. For experimentalists to invest the effort to understand and implement the technique they would first need to be convinced that is represents improvement over what is currently available.

Reviewer 2 Report

Title: Particle Size Inversion Constrained by L Norm for Dynamic Light Scattering

Manuscript ID: materials-1880766

Authors: Zhang et al.

Dear Authors,

Thank you for the opportunity to read your article. I found the topic is interesting and fundamental. Generally speaking, the methods and results were clearly presented while the discussion of the results need more addition with fair point of view and clear connections. I suggest that this article will be revised extensively before its re-submission for another review process if applicable. As a conclusion, I recommend its major revision at this state.

I hope my comments are helpful.

Good luck,

A reviewer

Major concerns:

“Keywords”

->Please consider providing keywords that are not used in the article title.

“1. Introduction”

-Please consider checking and if necessary revising the referencing. For example, your first reference starts from [14], not [1]. You also cite [14] for so many different things. Some numbers (e.g. 7, 10…13) are not in brackets.

-“The resulting DLS inversion is inherently complex and influenced by the different kind of noise due to fluctuations in the measurement duration, sample turbidity…to deviate significantly from the true distribution…to failure of particle size measurement.”->Please consider citing relevant references to support your statement.

-“…in this paper…a DLS inversion method with L2 model is proposed.”->Does this statement clearly reflect what you did in this study? If no, please consider revising it with an appropriate term for example Lp model?

-Based on your literature review, please consider clearly mentioning the unique contribution of this work.

“2. LP norm constrained regularization model for dynamic light scattering”

-“…we will choose p = 1…∞…to investigate the differences between different models and look for the optimal model.”->Please consider mentioning how you actually performed fitting data with p = ∞.

“3. Simulation experiment parameters and evaluation indicators”

-Equation 15 (and elsewhere): Please consider citing relevant reference(s).

-Equation 15: Please check and if necessary revise whether you wish to have u not µ.

-“…GCV 17, L-curve 18, and others.”->Please consider citing relevant references.

“4. Analysis of Simulated Data with Different Norms”

-In general, please discuss the results more. You will find some detail comments below.

-Table 1: Please consider providing the justification/reference of the parameters you selected.

-Figure 1 (and elsewhere): Please consider discussing the results in addition to their descriptions. For example, please consider discussing why “At the strong noise level of 0.01, distribution errors of L10-L has no big difference, while the distribution error of L2 is 0.1309…”

“5. Inversion analysis of L model”

“5.3. Discussion”

-Please try to incorporate and cite your results (e.g. numbers, figures & tables) in discussion. In the current statements of this section, the relationship between your results and discussion contents are not really clear to a reader.

“6. Experimental data inversion analysis”

-In general, please explain your experimental setup and procedure more. For example, you can inform the equipment and its producer, sample dilution procedure and vol.% of particles present in the particles, the application of sonication or not. Sample preparation and conditions could affect significantly the scattering experimental results.

https://www.lcpe.uni-sofia.bg/publications/2008/2008-09-PK-KD-ND-Handbook-Birdi-3rd-Edition.pdf

https://doi.org/10.1016/j.apt.2010.04.011

https://doi.org/10.3390/colloids2030037

-“The samples have been placed in a quartz cell with a diameter of 25nm…”->The samples with a diameter of 25 nm was placed in a quartz cell…? Where are the origin of 25 nm particles? You only had 38 nm particles and 55 nm+330 nm particles?

-How did you measure 38 nm particles and 55 nm+330 nm particles in addition to DLS? It could be useful to mention the equipment and procedure (and reference), as your analysis validation is solely relied on those numbers (i.e. 38 nm, 55 nm+330 nm).

-Please consider discussing the results.

“7. Conclusions”

-In this section, you may state some future perspectives.

Minor concerns:

-Please consider polishing English more. You may use some of my comments above and below for this purpose.

-“And the regularization scheme is related to…”->The regularization scheme is also related to…

-“…are used as an example to study…”->…were used as examples to study…”

-Please consider adding the line numbers throughout the article. That will help us easy to point out specific statements within the article for our discussions/communications.

Round 2

Reviewer 1 Report

The revision has addressed my concerns -- I recommend publication

Reviewer 2 Report

Dear Authors,

As all the comments were addressed, I would suggest the journal accept this article for its publication.

Best regards,
A reviewer